# The Intra-European Union Food Trade with the Relation to the Notifications in the Rapid Alert System for Food and Feed

**DOI:** 10.3390/ijerph18041623

**Published:** 2021-02-08

**Authors:** Marcin Pigłowski

**Affiliations:** Department of Quality Management, Faculty of Management and Quality Science, Gdynia Maritime University, Morska 81-87 Str., 81-225 Gdynia, Poland; m.piglowski@wpit.umg.edu.pl

**Keywords:** cluster analysis, European Union, food safety, food trade, RASFF

## Abstract

About three-quarters of food exports from European Union (EU) countries goes to the common market in which the free movement of products is ensured. Therefore, it is important to examine from which EU countries the food is exported, what food products they are, and what hazards may be present in these products. The data for research were obtained for 1999–2018 from the Eurostat database (according to the Standard International Trade Classification—SITC) and the Rapid Alert System for Food and Feed (RASFF) database. Then, cluster analysis was performed using joining (tree clustering) and two-way joining methods. The main food exporters were the following countries: Belgium, France, Germany, Italy, the Netherlands, Spain, and the United Kingdom. They exported: cereals, fruits and vegetables, beverages and feeding stuff (in quantitative terms) and fruits and vegetables, meat, dairy products, and cereals (in terms of value). In turn, the most frequently notified hazards in food originating from these countries were: pathogenic micro-organisms, microbial contaminants, metals, composition, foreign bodies, allergens, and pesticide residues. The increase in the number of alert notifications in the RASFF is particularly noticeable in recent years. The results of the research may be useful for activities related to food traceability, changes in the European law, and encouraging the use of extensive methods in agriculture.

## 1. Introduction

According to the Single European Act, the internal market is an area without borders with free movement of goods, persons, services, and capital [1,2,3]. Gasiorek et al. (2019) [4] believe that the lack of tariffs, origin checks, and regulatory checks at intra-European Union (EU) borders contribute to reducing transport times and lower transport costs (intra-EU refers to all transactions occurring within the EU and extra-EU related to transactions with all countries outside of the EU [5]). Miron (2018) [6] noted that EU trade policy was the first genuine common policy. In turn, Kropp et al. (2011) [7] reminded that the European Commission indicated transport and food sectors as critical infrastructure.

In the Regulation (EC) No 178/2002, food was defined as any substance or product, whether processed, partially processed, or unprocessed, intended to be, or reasonably expected to be ingested by humans. It includes drink, chewing gum, and any substance, including water, intentionally incorporated into the food during its manufacture, preparation, or treatment [8]. In turn, according to the Food and Agriculture Organization of the United Nations (FAO), food includes products in the Standard International Trade Classification (SITC) sections: 0 (food and live animals), 1 (beverages and tobacco), 4 (animal and vegetable oils, fats and waxes), and division 22 (oil-seeds and oleaginous fruits) [9,10].

In 2017, the EU was the first exporter and second importer (after the USA) of food and drinks in the world (19% share of global export and 12.6% of global import). But, what is important, in 2018, about 75% of EU food and drink exports were destined for the common market [11]. What is also important, according to the Eurostat (SITC), in the period 1999–2018, the share of intra-EU food trade in relation to extra-EU food trade increased by several percent (Table 1) [12].

Such a high level of intra-EU food trade may contribute to ensuring food security, i.e., the availability of food for the entire population [13]. In turn, according to the FAO, food security is a situation when all people, at all times, have physical, social, and economic access to sufficient, safe, and nutritious food that meets their dietary needs and food preferences for an active and healthy life [14]. In 2018, the largest ten importers and exporters within the intra-EU food and drink trade were Germany, the Netherlands, France, Belgium, Italy, Spain, Poland, the United Kingdom, Denmark, and Ireland, respectively [11]. Zolin and Uprasen (2018) [15] noticed that the first major traders in the intra-EU food trade in 1999–2015 were Germany, the Netherlands, and France. In turn, Alatriste-Contreras (2015) [16], highlighting that the food sector is one of the most traded within the EU, indicated the Netherlands and France. Urban et al. (2016) [17] also noticed that EU agricultural trade is relatively concentrated on intra-EU trade. Similarly, Faure (2018) [18] noted that food products are an important part of intra-EU trade and harmonization of food law contributed to its growth. Gandolfo (2014) [19] pointed out that share of intra-EU trade in total trade significantly increased, mainly due to agricultural products. Garman (2014) [20] noted that food trade is strongly growing and it is often an intra-industry type. In turn, Chevassus-Lozza and Galliano [21], referring to France, observed an increase in intra-firm food trade in the EU and the European multinational networks.

Intra-EU food trade is strengthened by the Common Agricultural Policy (CAP), i.e., set of legislation and practices adopted by the EU to provide common, unified policy on agriculture [13]. Urban et al. (2016) [17] noted that the EU is an outstanding example of heavily subsidized agriculture. Bakucs et al. (2019) [22] pointed out that one of the important targets of the CAP is to facilitate the spatial integration of agricultural markets within the individual countries, as well as throughout the EU. Garmann (2014) [20] reminded that the CAP accounts for more than 40% of the EU budget. However, it distorts trade. Despite many years of reforms in the field of the CAP to meet the requirements of the World Trade Organization (WTO), the EU is still criticized for supporting agricultural producers [17]. According to Swinbank (2017) [23], high import tariffs under the CAP protect EU producers, but complicate EU attempts to negotiate free trade agreements around the world. Sorgho and Larue (2014) [24] pointed out, however, that also the protection of indications of geographical origin of food products significantly affects trade between EU countries, because it can reduce information asymmetry between producers and consumers about product “quality”.

The food safety is freedom of food from anything harmful to human health [13]. According to food safety requirements in the Regulation (EC) No 178/2002, food shall not be placed on the market if it is unsafe (injurious to health or unfit for human consumption). An adverse health effect and the severity of this effect is related to the risk, which may result in a hazard (biological, chemical, or physical agent in food or condition of food) [8].

What is worrying, however, in the Rapid Alert System for Food and Feed (RASFF), the number of notifications regarding dangerous food derived from Europe is the highest, taking into account various regions of the world [25]. This system was created already in 1979, but currently the legal basis for it is the Regulation (EC) No 178/2002 [8]. The members of the RASFF are: EU countries, European Commission, the European Food Safety Authority (EFSA), the European Free Trade Association (EFTA) Surveillance Authority (ESA), Norway, Iceland, and Lichtenstein and also Switzerland. Thanks to the RASFF, they can quickly share information and react when the risks to public health are detected in the food chain. Alert information is sent when food or feed presenting serious risk is on the market and quick action may be required in another country than the notifying country (e.g., withdrawal or recall). The information notification relates to the food or feed in which the risk has been identified, but this does not require rapid action because the risk is not serious or the product is no longer on the market. In turn, border rejection may relate to a consignment of food or feed refused entry to the EU market due to a risk to human health, animal health, or the environment [25,26].

In 2017, the number of notifications in the RASFF for food from Europe was greater than for food from Asia (this happened for the first time since 2000) and in 2018 this difference was even greater [25]. What is more, in 2010–2018, until about 70% of all alert notifications and 40–70% of all information notifications in the RASFF related to food from EU countries [25,27,28,29,30,31,32,33].

Due to such a large and steadily growing food trade in the EU, it is important to recognize and track food hazards in the long run for countries and products where this trade is the highest. Therefore, the goal of this study was to identify EU countries with the intra-EU food trade above the mean value, products within this trade, and notifications in the RASFF in 1999–2018.

## 2. Materials and Methods

The data on intra-EU food trade (both export and import) in sections: 0—food and live animals (divisions 00–09), 1—beverages and tobacco (divisions 11, 12, and 19), 4—animal and vegetable oils, fats, and waxes (divisions 41, 42, 43, and 49), and division 22—oil-seeds and oleaginous fruits according to the SITC for the twenty years period from 1999 to 2018 was obtained from the Eurostat database [12]. Thus, such a long period also covered the years before the accession of Central and Eastern European countries to the EU in 2004, which made it possible to assess whether there were any changes in food trade in these countries after this year. The data from the Eurostat database included quantity in kg and value in euros and related to reporter country, partner country, year, and product. The data for previous years were usually not available, particularly for countries, which accessed EU in 2004 and after this year. However, for the examined period, the data were usually available and only few missing data were replaced with the number “0”. In the Eurostat database, there was a lack of data on food production for 1999–2018 or the data were inaccurate, therefore it was taken from the Faostat database [34]. In turn, the data on number of notifications related to the dangerous food were extracted from the RASFF database. This data covered notifying country, origin country, year, product, hazard, and notification type [26].

The names of data obtained from Eurostat and RASFF databases were shortened and/or modified. The data were processed using pivot tables, the transposition (when necessary), the mean value in Excel, and then the cluster analysis (joining-tree clustering and two-way joining) in Statistica 13.3 (TIBCO Software Inc.) was carried out. The tables and figures resulting from this analysis were included in Appendix A.

Cluster analysis means finding groups (clusters) of objects based on their similarity, so that within each cluster, the similarity of these objects is so great that the individual clusters are different enough from each other [35].

The purpose of joining cluster analysis was to join together objects into successively larger clusters. As a distance measure, the Euclidean distance was indicated. It is the geometric distance in the multidimensional space, most commonly chosen. The Ward’s method was indicated as linkage rule. It uses analysis of variance to evaluate the distances between clusters attempting to minimize the sum of squares of any two (hypothetical) clusters which can create at each step. This method is very efficient, but it tends to form clusters of small size. The results of joining cluster analysis were shown in vertical icicle plots.

The results of joining were supported by two-way joining cluster analysis. The first color of chart surface (showing a slight similarity) was adopted as white in order to increase the readability of charts. The next colors (from green, through yellow, orange, red to brown) indicated the increasing similarity. The two-way joining is used to uncover meaningful patterns of clusters. It could make difficulties in interpretation, because similarities between different clusters may relate to different subsets and finally clusters are not homogeneous. However, it is a useful tool for analyzing a large amount of data [36]. Two-way joining cluster analysis consisted in the simultaneous grouping of objects (as cases) and features (as variables). The objects were in the rows, and the features were in the columns of the tables (data matrix) from which the figures were generated. As a result of two-way joining cluster analysis, the rows and columns of tables are permuted in order to make clusters visible as blocks with objects, showing similar values across the features [37]. This method is particularly suited for analysis of very large data sets [38].

Depending on the examined aspect, data were placed in tables in different ways. For the data from the Eurostat database, they were: reporter country (exporting or importing), partner country (inversely: importing or exporting), year, and product. In turn, for the data from the RASFF database, they were: notifying country, origin country, year, product, hazard, and notification type.

## 3. Results

### 3.1. Intra-EU Food Trade According to the SITC

In the examined period (1999–2018), the intra-EU food trade according to the Eurostat (SITC) concerned mainly: cereals (23.0%), fruits and vegetables (19.6%), beverages (14.3%), and feeding stuff (11.8%), considering it quantitatively. In turn, the food trade in value related mainly to: fruits and vegetables (18.0%), meat (12.6%), dairy products (10.6%), and cereals (9.6%). It was assumed that the export and import values were approximately equal to each other. The percentage of all product sections according to the SITC was shown in Table 2.

The food trade in this period increased about two times, considering it quantitatively and about 2.5 times in value (Figure 1). Up to 2011, one kg of food subject to intra-EU trade cost less than one euro. Then, this trend reversed and a food price is still rising. However, in 2008–2009, there is also a visible drop in food prices related to the global economic crisis.

In this period, the percentage of food in total intra-EU export and import increased from 15% to 18% (in quantity) and from 9% to 11% (in euros) (Appendix A). The largest percentage share in intra-EU food exports in relation to its total intra-EU exports (in euros) had countries of Southern Europe (Cyprus, Greece, and Spain), Central, and Eastern Europe (Bulgaria, Latvia, Lithuania, and Poland), but also Denmark, France, Ireland, and the Netherlands. In turn, the smallest share in food exports in relation to its total intra-EU exports had other Central European countries, i.e., Czech Republic, Slovakia, and Slovenia and also Finland and Malta. It is worth also noting that the countries of Central and Eastern Europe had the greatest increase in intra-EU food exports in the analyzed period (particularly after their accession to the EU). Therefore, it can be concluded that these countries have benefited from joining the EU. On the other hand, these countries had also the highest food imports in relation to total intra-EU imports. Among the former EU member states, high imports had also Denmark, Greece, Ireland, Italy, Luxembourg, the Netherlands, Portugal, and the United Kingdom (Appendix A) [12]. It is also worth mentioning that some countries of Central and Eastern Europe (Bulgaria, Estonia, Latvia, Lithuania, Poland, and Romania) at least doubled their food production in 1999–2018 (Appendix A). However, among the countries with the food production above mean in this period were: France, Germany, Greece, Italy, the Netherlands, Poland, Romania, Spain, and the United Kingdom [34].

In turn, the mean value for the intra-EU trade, both for export and import in 1999–2018, was exceeded only in the case of seven Western European countries, i.e.,: Belgium, France, Germany, Italy, the Netherlands, Spain, and the United Kingdom (Table 3).

Thus, they were the countries with the highest food production in the EU (Belgium was the only additional country). In the case of these countries, high exports and imports resulted not only from the concentration of food production in this part of Europe, but also due to cooperation in the food industry, a dense network of communication routes and experience in meeting food quality requirements related to law or standards.

Each of the Western European countries above mentioned (Belgium, France, Germany, Italy, the Netherlands, Spain, and the United Kingdom) were responsible for a few to several percent of intra-EU food trade, but together they covered 75% of this trade. In the examined period, the largest food turnover (considering both billions of kg and billions of euros) had Germany and then France and the Netherlands. However, the surplus in the total trade balance within food (in billions of euros) had the Netherlands (the highest value) and then Spain, Belgium, and France. In turn, trade deficit can be observed in the case of Italy, Germany, and the United Kingdom (the lowest value).

Considering total food intra-EU export in 1999–2018, the Netherlands, France, Germany, Spain, Italy, and Belgium created a separate cluster in the joining cluster analysis (left side of Appendix A). In turn, in the case of total food intra-EU import, all of the discussed countries created a separate cluster (left side of Appendix A). The other countries formed a cluster with subclusters on the right side of Appendix A. However, the largest food trade can be observed between the Netherlands and Germany (Appendix A).

Table 4 presented results of two-way joining cluster analysis within intra-EU food trade (countries and products are arranged in alphabetical order), with the indication of clusters in appropriate figures.

Table 4 was based on Appendix A and concerned the food trade between Belgium, France, Germany, Italy, the Netherlands, Spain, and the United Kingdom, respectively, and other EU countries. The highest values (brown, red, and orange colors) related both to particular years and products usually concentrated at the bottom of the charts and mainly referred to trade between the above-mentioned countries. What is important, in the analyzed period, the largest trade occurred usually in the recent years, particularly since 2008/2009 (panel “a” of Appendix A), regardless of the country discussed. In turn, products were presented in panel “b” of these figures. However, considering partner countries for particular reporter countries in question, one can observe that food trade related to years and products do not always overlaps. It is a result of varied food trade structures for individual reporter countries, as well as that clusters may not be homogeneous—as mentioned in Data and Methods.

#### 3.1.1. Export

Beverages, in terms of quantity, were mainly exported by Germany to the Netherlands and by Italy to France. However, given the highest values in euros, beverages were exported by France to Belgium, Germany, and the United Kingdom, by Italy to Germany, and by the United Kingdom to France and Spain.

Cereals represented the largest quantitatively share in export from Belgium to the Netherlands, from France to Belgium, Italy, the Netherlands, and Spain, and from the United Kingdom to Ireland and Spain. In turn, when consider cereals in value, they accounted for the largest share in export from France to Belgium and from the United Kingdom to Ireland.

Feeding stuff accounted for the important quantitative share in export from Belgium to France and the Netherlands and from the Netherlands to Germany.

In turn, fruits and vegetables, in terms of quantity, were mainly exported from Belgium to France, Germany, and the Netherlands, from the Netherlands to Germany, and from Spain to France and Germany. Similarly, in terms of value in euros, the most important trade partners for Belgium within fruits and vegetables were France, Germany, and the Netherlands, for Italy and the Netherlands, it was Germany, and for Spain, there were France and Germany.

Taking into account the value in euros, they were also other food products exported, i.e.,

dairy products and birds’ eggs from Germany to Italy and the Netherlands and from the United Kingdom to Ireland,fish and seafood from the United Kingdom to France,live animals from France to Italy,meat from Germany to Italy and the Netherlands and from the United Kingdom to Ireland.

#### 3.1.2. Import

In the case of import, the major trading partners for particular countries could be other than in the case of export. When considering the quantity (in kg), beverages were imported mainly by Belgium from the Netherlands, by France from Italy, and by the Netherlands from Germany. However, taking into account the value in euro, they were imported mainly by Belgium and the United Kingdom from France.

For five countries (Belgium, Italy, the Netherlands, Spain and the United Kingdom) the most important partner within cereals in terms of quantity was France. In turn, taking into account value in euros, cereals was mainly imported by Belgium and Spain from France and by the Netherlands from France and Germany.

Feeding stuff (taking into account the quantity) was mainly imported by Germany from the Netherlands.

In turn, fruits and vegetables, in terms of quantity, were imported mainly by France from Spain and by Germany and the United Kingdom from the Netherlands and Spain. However, in the case of value in euros, fruits and vegetables were imported by Belgium from France and the Netherlands, by France from Spain, by Germany and the United Kingdom from the Netherlands and Spain, by the Netherlands from Belgium, Germany and Spain and by Spain from France.

Taking into account the value in euros there were also other food products imported:dairy products and birds’ eggs—by Belgium from France and the Netherlands, by Italy and the Netherlands from Germany, by Spain from France,live animals—by Italy from France,meat—by Italy and the Netherlands from Germany, by the United Kingdom from Ireland and the Netherlands,miscellaneous products—by the Netherlands from Belgium and Germany,tobacco—by Spain from Germany.

### 3.2. Notifications on Food from the EU in the RASFF

In the period considered (1999–2018), there were 19,601 notifications in the RASFF on products originated from EU countries. The most frequently notified were: fish (13.8%), meat (11.0%), fruits and vegetables (9.2%), poultry meat (7.7%), dietetic foods (5.9%), cereals (5.8%), feed materials (5.1%), and milk (4.5%) (Table 5). It is worth noting, however, that within the RASFF, there are several times more product categories than in the SITC classification (Table 2), which is why these percentages are more dispersed. The total number of notifications generally raised last years and in 2018 exceeded 1600. The number of alert notifications this year was higher than information notifications (Figure 2).

The two-way joining cluster analysis confirmed the increase of notifications in the last years (Appendix A) and notification types (S37b). However, most frequently notified hazards (with the value above the mean) in the examined period were: pathogenic micro-organisms (21.3%), microbial contaminants (12.4%), metals (7.7%), composition (6.8%), foreign bodies (6.6%), allergens (5.4%), pesticide residues (5.1%), food additives (5.0%), and mycotoxins (4.8%).

#### 3.2.1. The General Results

The notified products originated mainly from the same countries that are the main food exporters within the EU (Appendix A). Notification to products from Belgium, France, Germany, Italy, the Netherlands, Spain, the United Kingdom, and additionally also from Poland accounted for over 68% of all notifications for products originating from EU countries). The number of notifications in the RASFF for products originating from these eight countries exceeded the mean value. They created the separate cluster with subclusters in the joining cluster analysis. However, the similarities in notifications within these countries are significantly differentiated (the left side of panel “a” in Appendix A). In turn, other EU countries created a cluster on the right side of these panels.

Considering the entire EU using two-way joining cluster analysis, the largest number of notifications was reported by Germany and Italy for products originating from these countries, respectively. Italy notified also products from Spain (Appendix A). A problem that was noticeable across the EU were metals (Appendix A) in fish originating from Spain (Appendix A). A visible hazard was also pathogenic micro-organisms in products from France, Germany, the Netherlands, and Poland (Appendix A).

#### 3.2.2. The Detailed Results

The detailed results of two-way joining cluster analysis related to notifications on food in the RASFF were presented in Table 6 (countries, years, products, and hazards are arranged in alphabetical order). This table indicated clusters visible in appropriate figures (orange, red, and brown colors in Appendix A). The analysis was carried out for previously examined countries, i.e., Belgium, France, Germany, Italy, the Netherlands, Spain, and the United Kingdom, taking into account: year (panel “a” of the mentioned figures), product (panel “b”), hazard (panel “c”), notification type (panel “d”), and notifying country. As in the case of the analysis for food trade, the results for the notifying country do not always overlap, because individual clusters may not be homogeneous. However, it can be seen that in most cases, the origin country coincides with the notifying country and related to the last years with alert and information as a notification type, which was already signaled before.

Taking into consideration the clusters occurring both within products and hazards, the most important attention should be paid to:bivalve molluscs (microbial contaminants) from Italy notified by this country,cereals (foreign bodies and/or allergens) from Germany and the United Kingdom reported by these countries,fish (metals) from Spain reported by Italy,fruits and vegetables (pesticide residues) from Italy notified by Germany,meat (microbiological contaminants and/or pathogenic micro-organisms) from Belgium, Germany, and Italy notified by these countries, products from Germany was also reported by Denmark,milk (microbiological contaminants and/or pathogenic micro-organisms) from France notified by this country,poultry meat (microbiological contaminants and/or pathogenic micro-organisms) from Belgium and notified by this country and from Germany reported by Denmark.

Within the individual hazard categories, the following specific hazards were most frequently notified:allergens (celery, eggs, gluten, lactoprotein, lactose, milk, mustard, nuts, sesame, soya, and wheat),foreign bodies (pieces of bones, dead insects and mice, glass, metal, plastic, rubber, small stones, and wood),metals (arsenic, cadmium, chromium, lead, mercury, nickel, and tin),microbial contaminants (mainly non-pathogenic Bacillus spp., Enterobacteriaceae, Escherichia coli, Listeria monocytogenes, Pseudomonas spp., Salmonella spp., and also molds and yeasts),pathogenic micro-organisms (Bacillus cereus, Campylobacter spp., Clostridium spp., Listeria monocytogenes, Salmonella spp. and also hepatitis A, histamine, and norovirus),pesticide residues (e.g., carbofuran, chlorate, chlormequat, chlorpyrifos, dichlorvos, dimethoate, dithiocarbamates, endosulfan, ethephon, fipronil, isofenphos-methyl, nitrofen, and oxamyl) [26].

## 4. Discussion

### 4.1. Intra-EU Food Trade. Viewpoint of Various Authors

#### 4.1.1. Western European Countries

Bermejo (2014) [39] highlighted that the food chain is formed by production, transport, preservation, and distribution. In turn, the footprint of transport depends on the distance, mode used, and kind of food. The greatest impact on the environment includes dairy farming, cattle farming, grain crop production, and fisheries. These kinds of foods are, unfortunately, among which are often the subject of intra-EU trade (Table 2). What is also important, Pelletier et al. (2018) [40] indicated these and other products as having the greatest impact on social risk (the potential of exposure to negative social conditions that undermine social sustainability), taking into account the country of origin in terms of intra-EU imports. In the aspect of social risk among food items they mentioned, e.g., dairy products, bovine cattle, cereals, fish, but also vegetables, fruits, and nuts, beverages and tobacco, sugar cane and sugar beet. In turn, Kleter et al. (2018) [41] noted that a significant amount of feed materials is also traded between EU countries but also comes from outside the EU, which causes difficulty in tracking them.

As already previously mentioned, during the period considered, the food trade mainly took place between seven countries of the western part of the EU, i.e., Belgium, France, Germany, Italy, the Netherlands, Spain, and the United Kingdom. Ireland was the only additional country due to close trade relations with the United Kingdom. All of these countries (except the United Kingdom) were the founders of the European Monetary Union (EMU). Huchet-Bourdon and Cheptea (2011) [42] established that food trade between EMU countries is sensitive to the quality and similarity of institutions as well as the availability of information about foreign partners. They added, however, that this impact is smaller if a large amount of information is exchanged and the institutional framework is strong.

The results of studies on intra-EU food trade carried out by various authors are consistent with those presented in Table 4. Quested et al. (2010) [43] noted that the EU is self-sufficient in most type of food types (e.g., meat), but this self-sufficiency varies from country to country. However, this makes intra-EU trade more important for European countries than extra-EU trade. In turn, Emmanouilides and Fousekis (2015) [44] pointed out that the EU is a major producer and consumer of chicken meat at the global scale, but more than 75% of this product is traded within the EU. Ghazalian et al. (2011) [45] noted that preferential tariffs were the main factor that contributed to the increase in intra-EU meat trade. Karamera et al. (2015) [46] also pointed out that common borders stimulate meat trade among EU countries. However, Poppy et al. (2019) [47] noted also that the Netherlands imports a lot of meat from non-EU countries via Rotterdam.

Bejnec and Fertő (2014) [48] pointed out that the EU dairy market is very protected from foreign competition. They added that it leads to a greater export of dairy products to the internal market than the external one, but also causes strong competition within the EU. Philippidis and Waschik (2019) [49] noted that the abolition of milk quotas in 2015 will cause even greater increase in intra-EU dairy trade in the future.

Some kinds of fresh fruits and vegetables are another example of EU protection against foreign competition. The EU has introduced an entry price system (EPS) that restricts imports below the product-specific and politically determined entry price level [50]. This can be used by EU countries that have a temperate and Mediterranean climate. In 2015, three EU countries accounted for more two-thirds of intra-EU exports of fruits and vegetables, i.e., Spain (exporter of most fruits and vegetables), Italy (grapes and apples), and the Netherlands (tomatoes) [51].

#### 4.1.2. Potential Effects of Brexit

Many authors highlighted the potential consequences of Brexit, referring them to the intra-EU food trade. Cheptea and Huchet (2019) [52] and Swinbank (2016) [53] noted that the main trading partners for the United Kingdom in terms of food are EU countries. In 2015, the main products, which were imported from the EU to the United Kingdom, were wines, bakery products, chocolate, cheese, and meat [52]. Poppy et al. (2019) [47] also noted that the vast majority of meat (beef and pork) as well as poultry imported into the United Kingdom originate from EU countries, mainly from Denmark, Germany, Ireland, the Netherlands, and Poland. However, Poppy et al. (2019) [47] and Swinbank (2016) [53] noticed that when it comes to the Netherlands, some imports from this country to the United Kingdom come also from other sources via the port of Rotterdam, as mentioned before. Therefore, Poppy et al. (2019) [47] pointed out that the United Kingdom should conclude trade agreements (regarding meat imports) directly with these countries. However, Swinbank (2017) [23] noted that import tariffs maintained under the CAP will be a problem for the United Kingdom after Brexit, especially on the border with Ireland. Cheptea and Huchet (2019) [52] and Matthews (2017) [54] have just noticed that the Irish agri-food sector in particular will be exposed to losses. Jacobs (2018) [55] pointed out that Ireland would lose the major market for its beef and dairy products, and the Netherlands and Denmark for their pork. Van Berkum et al. (2018) [56] noted that the British market accounts for ten percent of Dutch agricultural exports. But for Germany, food and live animals was only the fifth export category to the United Kingdom in 2016 [57].

In turn, Swinbank (2014) [58] pointed out that the loss of the major net contributor (which is the United Kingdom) to the EU budget can cause changes in the CAP. In fact, the CAP was a particular object of criticism from the United Kingdom [55,59], which proposed 50% cuts in this policy [59]. Matthews (2016) [60] noted that the United Kingdom was not only a net contributor, but also a net importer of agri-food products from the EU. Hubbard et al. (2018) [61] stated that the British agri-food sector will be severely affected by Brexit because it is just subsidized and regulated under the CAP and it is also dependent on migrant labor. According to Gasiorek et al. (2019) [4], there will be a significant decline in exports and imports in the food sector after Brexit. In turn, Poppy et al. (2019) [47] believed that closing gaps resulting from imports into the United Kingdom from the EU may cause increase in prices. However, Boulanger and Philippidis (2015a) [62] noted that the withdrawal of the United Kingdom from the CAP would be ultimately beneficial for this country.

#### 4.1.3. Eastern European Countries

In the context of intra-EU food trade, it is also worth mentioning the Eastern European countries that joined the EU in 2004. Zolin and Uprasen (2018) [15] noticed that after the EU enlargement increased intra-EU food trade for the following products: beverages and tobacco, cereals, fruits and vegetables, seafood, sugars and also animal and vegetable oils, feedstuffs and live animals. Bejnec and Fertő (2008, 2009a, 2012) [63,64,65] also pointed out that the EU enlargement contributed to the increase in food trade.

Chevassus-Lozza et al. (2008) [66] noted that despite the increase in exports of agri-food products from these countries to EU countries already in the pre-accession period (1999–2004), they faced impediments to trade, mainly due to tariffs and sanitary requirements. According to Jambor (2014) [67], although the share of trade in agricultural products increased significantly after the accession of new countries to the EU, these products were of low quality and complementary rather than competitive.

In turn, Bojnec and Fertő (2009b) [68] showed that already in 1995–2003, Poland and Hungary have caught up in successful quality competition and to a lesser extent also in successful price competition in the field of agro-food trade with EU countries. Antohi et al. (2019) [69] noted that agri-food trade became even a specialization of the Visegrad countries (Poland, the Czech Republic, Slovakia, and Hungary) after their accession to the EU, although initially these countries faced a deterioration of trade balance in this respect. Bejnec and Fertő (2009a) [64], Csaki (2016) [70], and Kocsis and Major (2018) [71] noticed that particularly Poland significantly benefited from the opening borders.

### 4.2. Food from the EU in RASFF and Other Reports

Due to the different names and the number of product categories according to the SITC and the RASFF, it is difficult to compare them. However, the products that were most often the subject of intra-EU food trade were usually also most often reported in the RASFF, e.g., cereals, dairy products, fruits and vegetables, meat (including poultry), and fish. Table 7 presented the values of the Pearson’s correlation coefficient for the amount of food in billions of kg exported from individual countries to other EU countries (according to the SITC) and the number of notifications on food originating from these countries (according to the RASFF, with the exception of notifications from the origin countries). The value of this coefficient was 0.82 for all EU countries, 0.81 for Western EU countries (in both cases high correlation), and only 0.63 for Eastern EU countries (moderate correlation). In each examined group of countries, the value of the test statistic exceeded the value of the critical statistic in two-tailed distribution, therefore these associations are statistically significant [72].

Similar values of the correlation coefficient for all EU and Western EU countries confirm their advantage over the Eastern EU countries in the intra-EU trade in food and economic integration in this area. There are, however, many potential factors that could affect intra-EU food trade and notifications in the RASFF. Houghton et al. (2008) [73] noted that the introduction of strict standards, quality control, and monitoring procedures in the food chain in Western EU countries in the years before 2008 resulted in an increase in the number of alerts (which was called the “the paradox of progress”). However, another effect can be also noticed: the introduction of border rejections [8], i.e., rejections at the external EU border since 2008 caused, in turn, a decrease in the number of alerts (Figure 2). What is also important, if all notifications in the RASFF are taken into account, hazards in food were most often reported by Germany, Italy, Spain, and the United Kingdom [74,75]. Pigłowski (2017) [76] added also France and the Netherlands. All these countries had the intra-EU food trade above the mean value in the EU in the analyzed period. However, apart from legal requirements, the number of notifications in the RASFF may also be influenced by other factors, e.g., growing share of intra-EU food trade compared to extra-EU food trade (Table 1), the increasingly industrial food production process (which may increase the number of chemical and animal diseases hazards), the structure of food production and distribution in connection with geographical conditions, the development of the transport network, the distances and condition of the means of transport, and also the development of a network of food control authorities and the experience of their employees.

In the RASFF annual reports, the top ten number of notifications (according to hazard type and product category) by origin country usually include products from Asia. However, they more and more often also concern EU countries, i.e., norovirus in bivalve molluscs from France [25], fipronil in eggs from Italy [33], carbon monoxide in fish from Spain [29], mercury in fish from Spain [26,27,29,30,31,32,33], Listeria in fish from Poland [29], and Salmonella in poultry from Poland [25,29,33]. Especially disturbing are notifications repeated in many reports relating to the presence of mercury (heavy metal) in fish in Spain. This hazard was so often notified that it was noticeable after taking into account notifications from the entire period under investigation as mentioned before (Appendix A).

RASFF notifications that relate only to food originating from the EU were rarely noticed by other authors. However, D’Amico et al. (2018) [77] also noted that the highest number of RASFF notifications on seafood in 2011–2015 related to products originating from Spain, but also from Italy. Whereas Noël et al. (2011) [78] noticed that in 2007–2009, Italy notified cadmium in crabs originating from France, Ireland, and the United Kingdom. Petrović and D’Agostino (2016) [79] drew attention to the RASFF notifications regarding norovirus in oysters from France in 2009–2014 and in raspberries from Poland and Serbia in 2012–2014. De Keuckelaere et al. (2015) [80] indicated notifications on norovirus in frozen berries from Poland in the similar period. In turn, Lüth et al. (2019) [81] conducted an analysis of notifications regarding the presence of Listeria monocytogenes in products on the German market in 2001–2015. They stated that these notifications mostly concerned milk and milk products, fish and fish products, meat and meat products (other than poultry) originating from France, Germany, Italy, and Poland. However, Brandão et al. (2015) [82] noticed that Salmonella enterica was the most common cause of notifications food in Europe in 2012–2014.

## 5. Conclusions

In 1999–2018, the intra-EU food trade according to the Standard International Trade Classification (SITC) concerned mainly cereals, fruits and vegetables, beverages, and feeding stuff (in total approx. 70% in quantitative terms). In turn, the food trade in value related mainly to fruits and vegetables, meat, dairy products, and cereals (about 50% in total). The relation of quantity to value allows to state that the price of food was constantly rising. Three quarters of intra-EU food trade was conducted between Western European countries, i.e., Belgium, France, Germany, Italy, the Netherlands, Spain, and the United Kingdom. It indicates a high level of economic integration between aforementioned countries in this area.

The most commonly reported food hazards (mainly as alerts) notified in the Rapid Alert System for Food and Feed (RASFF) were pathogenic micro-organisms and microbiological contaminants (in meat, poultry meat, and milk), microbial contaminants (in bivalve molluscs), metals (in fish), foreign bodies and allergens (in cereals), and pesticide residues (in fruits and vegetables). These hazards were mostly notified by the countries from which the food originated, which means a high level of awareness about food hazards in the supervisory authorities in individual countries. It is also the result of the obligation to comply with EU law in this area. However, it was also a high correlation between the amount of food in billions of kg exported from individual countries to other EU countries and the number of notifications on food from these countries reported by other EU countries. In the context of the single market and the free movement of products (including re-export), the traceability aspect is particularly important, as it allows eliminating or reducing hazards in food, as well as improving the EU law.

The Common Agricultural Policy (CAP) and the EU protection against foreign competition contribute to strengthening intra-EU food trade and increasing food security. The opening of this market could increase food exports from the EU, but would result in the need to control imported food in terms of its safety. The important issue is also Brexit. Due to the fact that the United Kingdom is a significant importer of food from the EU, it is important to maintain this market. It is therefore necessary to conclude an appropriate trade agreement, beneficial to both parties, and taking into account the requirements of food safety. The Central and Eastern European countries that have joined the EU have seen significant increases in their food production and intra-EU food trade. They may constitute competition in this respect for the Western EU countries due to the more extensive way of farming. This could contribute to a more balanced development of these countries and increase of food safety.

The main difficulties in the research were missing data in the Eurostat and RASFF databases, as well as different product names in these bases. Further research directions could concern the correlation between food production (using the Faostat database) and intra- and extra-EU food trade. This should also be connected to the notifications in the RASFF. Such studies would more accurately assess the food security of the EU (within particularly product categories), but also its food safety in an international context.

## Figures and Tables

**Figure 1 ijerph-18-01623-f001:**
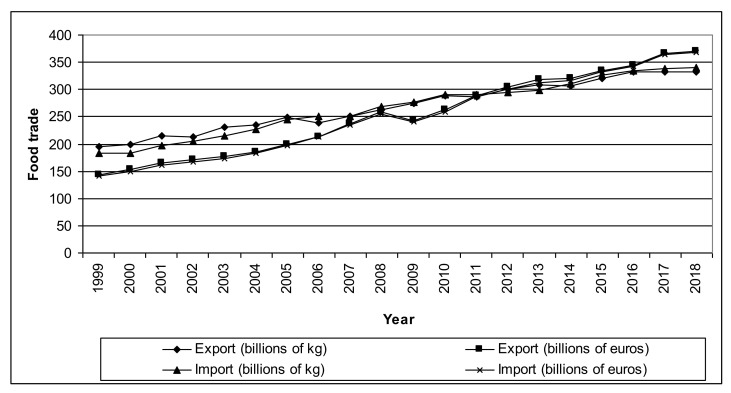
Intra-EU food trade in 1999–2018 according to the Eurostat (SITC).

**Figure 2 ijerph-18-01623-f002:**
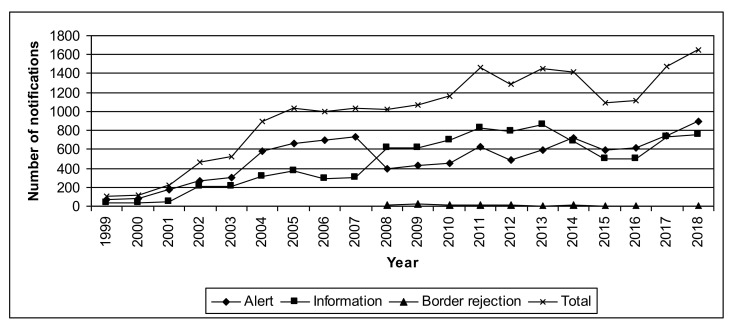
Number of notifications on food originated from EU countries in 1999–2018 according to the RASFF.

**Table 1 ijerph-18-01623-t001:** Percentage of intra-EU and extra-EU food trade according to the Eurostat (the Standard International Trade Classification SITC).

Year	Intra-EU Trade				Extra-EU Trade			
	Export		Import		Export		Import	
	(kg)	(Euros)	(kg)	(Euros)	(kg)	(Euros)	(kg)	(Euros)
1999	73.0%	79.5%	63.0%	71.5%	27.0%	20.5%	37.0%	28.5%
2000	73.9%	79.1%	63.9%	71.3%	26.1%	20.9%	36.1%	28.7%
2001	78.0%	79.9%	62.4%	71.2%	22.0%	20.1%	37.6%	28.8%
2002	77.5%	80.4%	61.5%	71.7%	22.5%	19.6%	38.5%	28.3%
2003	77.7%	81.7%	62.8%	72.7%	22.3%	18.3%	37.2%	27.3%
2004	81.0%	82.2%	63.6%	73.4%	19.0%	17.8%	36.4%	26.6%
2005	79.8%	82.6%	65.8%	73.4%	20.2%	17.4%	34.2%	26.6%
2006	79.8%	82.4%	66.1%	73.5%	20.2%	17.6%	33.9%	26.5%
2007	82.3%	82.8%	64.3%	73.2%	17.7%	17.2%	35.7%	26.8%
2008	79.9%	82.0%	64.9%	72.7%	20.1%	18.0%	35.1%	27.3%
2009	80.6%	82.5%	69.5%	74.2%	19.4%	17.5%	30.5%	25.8%
2010	78.5%	80.6%	70.5%	73.8%	21.5%	19.4%	29.5%	26.2%
2011	78.6%	79.3%	69.0%	72.9%	21.4%	20.7%	31.0%	27.1%
2012	79.0%	78.6%	69.6%	73.4%	21.0%	21.4%	30.4%	26.6%
2013	76.1%	78.2%	69.5%	74.2%	23.9%	21.8%	30.5%	25.8%
2014	75.3%	77.8%	69.4%	73.8%	24.7%	22.2%	30.6%	26.2%
2015	74.3%	77.7%	70.2%	72.9%	25.7%	22.3%	29.8%	27.1%
2016	74.6%	77.5%	70.5%	73.4%	25.4%	22.5%	29.5%	26.6%
2017	77.3%	77.9%	69.7%	73.9%	22.7%	22.1%	30.3%	26.1%
2018	77.7%	78.2%	68.6%	74.2%	22.3%	21.8%	31.4%	25.8%

**Table 2 ijerph-18-01623-t002:** Intra-EU food trade in 1999–2018 for particular products according to the Eurostat (SITC).

Product (SITC Code)	Percentage (kg)	Percentage (Euros)
animal oils, fats (41)	0.5%	0.4%
animal/vegetable fats, oils (43)	0.8%	0.6%
animal/vegetable oils-other (49)	below 0.1%	0.1%
beverages (11)	14.3%	8.8%
beverages/tobacco-other (19)	0.1%	0.3%
cereals (04)	23.0%	9.6%
cocoa, coffee, tea, spices (07)	1.7%	6.4%
dairy products, birds’ eggs (02)	7.2%	10.6%
feeding stuff (08)	11.8%	4.6%
fish, seafood (03)	1.6%	6.2%
fixed vegetable fats, oils (42)	2.9%	3.1%
fruits, vegetables (05)	19.6%	18.0%
live animals (00)	1.1%	2.4%
meat (01)	4.6%	12.6%
miscellaneous (09)	2.9%	8.1%
oil seeds, oleaginous fruits (22)	4.0%	1.7%
sugars, honey (06)	3.7%	2.8%
tobacco (12)	0.3%	3.8%

**Table 3 ijerph-18-01623-t003:** Intra-EU food trade in 1999–2018 for individual countries according to the Eurostat (SITC).

Reporter Country	Export				Import			
	Billions of kg	Percentage	Billions of Euros	Percentage	Billions of kg	Percentage	Billions of Euros	Percentage
Austria	125.3	2.3%	120.8	2.4%	131.8	2.5%	142.2	2.8%
Belgium	449.2	8.4%	469.5	9.3%	590.5	11.1%	369.6	7.4%
Bulgaria	68.1	1.3%	29.0	0.6%	22.6	0.4%	24.0	0.5%
Croatia	37.8	0.7%	9.7	0.2%	28.0	0.5%	24.2	0.5%
Cyprus	3.2	0.1%	2.8	0.1%	20.5	0.4%	11.9	0.2%
Czech Republic	196.2	3.7%	73.4	1.5%	91.3	1.7%	88.9	1.8%
Denmark	121.9	2.3%	191.5	3.8%	102.2	1.9%	114.6	2.3%
Estonia	12.6	0.2%	10.4	0.2%	15.1	0.3%	17.6	0.4%
Finland	15.1	0.3%	13.5	0.3%	40.9	0.8%	55.5	1.1%
France	915.8	17.0%	640.2	12.7%	534.0	10.0%	595.1	11.9%
Germany	902.4	16.8%	775.3	15.3%	951.4	17.9%	919.4	18.4%
Greece	63.8	1.2%	61.8	1.2%	76.0	1.4%	90.6	1.8%
Hungary	173.0	3.2%	85.9	1.7%	57.6	1.1%	56.8	1.1%
Ireland	57.0	1.1%	132.4	2.6%	92.0	1.7%	99.3	2.0%
Italy	359.3	6.7%	355.9	7.0%	431.2	8.1%	472.8	9.5%
Latvia	24.8	0.5%	13.7	0.3%	25.4	0.5%	24.1	0.5%
Lithuania	39.2	0.7%	30.5	0.6%	34.1	0.6%	33.3	0.7%
Luxemburg	12.9	0.2%	16.5	0.3%	18.1	0.3%	33.2	0.7%
Malta	0.4	0.0%	0.4	0.0%	7.0	0.1%	8.0	0.2%
Netherlands	723.5	13.5%	883.0	17.5%	670.7	12.6%	437.7	8.8%
Poland	176.5	3.3%	204.9	4.1%	247.0	4.6%	148.8	3.0%
Portugal	46.8	0.9%	57.2	1.1%	132.7	2.5%	113.3	2.3%
Romania	73.6	1.4%	37.5	0.7%	69.6	1.3%	55.0	1.1%
Slovakia	55.9	1.0%	34.7	0.7%	50.9	1.0%	49.1	1.0%
Slovenia	66.4	1.2%	14.9	0.3%	42.3	0.8%	24.3	0.5%
Spain	402.4	7.5%	460.7	9.1%	324.5	6.1%	295.6	5.9%
Sweden	50.0	0.9%	75.9	1.5%	91.9	1.7%	124.5	2.5%
United Kingdom	199.9	3.7%	249.5	4.9%	425.8	8.0%	562.2	11.3%
**Mean value**	191.9		180.4		190.2		178.3	

**Table 4 ijerph-18-01623-t004:** Results of two-way joining cluster analysis within intra-EU food trade.

Reporter Country	Flow	Product (Partner Country)	
		by Quantity (kg)	by Value (Euros)
Belgium	export	cereals (Netherlands), feeding stuff (France, Netherlands), fruits, vegetables (France, Germany, Netherlands); Appendix A	fruits, vegetables (France, Germany, Netherlands); Appendix A
	import	beverages (Netherlands), cereals (France); Appendix A	beverages, cereals (France), dairy products, birds’ eggs, fruits, vegetables (France, Netherlands); Appendix A
France	export	cereals (Belgium, Italy, Netherlands, Spain); Appendix A	beverages (Belgium, Germany, United Kingdom), cereals (Belgium), live animals (Italy); Appendix A
	import	beverages (Italy), fruits, vegetables (Spain); Appendix A	fruits, vegetables (Spain); Appendix A
Germany	export	beverages (Netherlands); Appendix A	dairy products, birds’ eggs, meat (Italy, Netherlands); Appendix A
	import	feeding stuff (Netherlands), fruits, vegetables (Netherlands, Spain); Appendix A	fruits, vegetables (Netherlands, Spain); Appendix A
Italy	export	beverages (France); Appendix A	beverages, fruits, vegetables (Germany); Appendix A
	import	cereals (France); Appendix A	dairy products, birds’ eggs, meat (Germany), live animals (France); Appendix A
Netherlands	export	feeding stuff, fruits, vegetables (Germany); Appendix A	fruits, vegetables (Germany); Appendix A
	import	beverages (Germany), cereals (France); Appendix A	cereals (France, Germany), dairy products, birds’ eggs (Germany), fruits, vegetables (Belgium, Germany, Spain), meat (Germany), miscellaneous (Belgium, Germany); Appendix A
Spain	export	fruits, vegetables (France, Germany); Appendix A	fruits, vegetables (France, Germany); Appendix A
	import	cereals (France); Appendix A	cereals, dairy products, birds’ eggs, fruits, vegetables (France), tobacco (Germany); Appendix A
United Kingdom	export	cereals (Ireland, Spain); Appendix A	beverages (France, Spain), cereals, dairy products, birds’ eggs, meat (Ireland), fish, seafood (France); Appendix A
	import	cereals (France), fruits, vegetables (Netherlands, Spain); Appendix A	beverages (France), fruits, vegetables (Netherlands, Spain), meat (Ireland, Netherlands); Appendix A

**Table 5 ijerph-18-01623-t005:** Notifications in the Rapid Alert System for Food and Feed (RASFF) in 1999–2018 for particular products originated from EU countries.

Product	Percentage	Product	Percentage
alcoholic beverages	0.4%	food contact materials	3.6%
animal by-products	0.1%	fruits, vegetables	9.2%
animal nutrition	1.5%	gastropods	0.1%
bivalve molluscs	3.8%	herbs, spices	3.3%
cephalopods	0.3%	honey	0.7%
cereals	5.8%	ices, desserts	0.5%
cocoa, coffee, tea	1.9%	meat	11.0%
compound feeds	0.8%	milk	4.5%
confectionery	1.9%	molluscs	0.7%
crustaceans	2.1%	natural mineral water	0.2%
dietetic foods	5.9%	non-alcoholic beverages	1.0%
eggs	1.8%	nuts, seeds	3.6%
farmed crustaceans	below 0.1%	other food product	0.8%
farmed fish	below 0.1%	pet food	1.1%
fats and oils	1.0%	poultry meat	7.7%
feed additives	0.2%	prepared dishes	2.3%
feed for food	0.7%	soups, broths	1.8%
feed materials	5.1%	water for human	0.1%
feed premixtures	0.2%	wild caught fish	0.1%
fish	13.8%	wine	0.1%
food additives	0.2%		

**Table 6 ijerph-18-01623-t006:** Results of two-way joining cluster analysis related to notifications on food in the RASFF.

Origin Country	Year	Product	Hazard	Notification Type
	(Notifying Country)			
Belgium	2017, 2018 (Belgium); Appendix A	meat, poultry meat (Belgium); Appendix A	microbial contaminants, pathogenic micro-organisms (Belgium); Appendix A	alert (Belgium); Appendix A
France	2014–2018 (France), 2005 (Italy); Appendix A	crustaceans, fish (Italy), milk (France); Appendix A	microbial contaminants, pathogenic micro-organisms (France); Appendix A	alert (France), information (Italy); Appendix A
Germany	2011 (Denmark), 2002–2005, 2011–2012, 2014, 2016–2018 (Germany); Appendix A	cereals, dietetic food, meat (Germany), poultry meat, meat (Denmark); Appendix A	foreign bodies, pathogenic micro-organisms (Germany), pathogenic micro-organisms (Denmark); Appendix A	alert, information (Germany); Appendix A
Italy	2002 (Germany), 2017, 2018 (Italy); Appendix A	bivalve molluscs, meat (Italy), fruits, vegetables (Germany); Appendix A	microbial contaminants (Italy), pesticide residues (Germany); Appendix A	alert, information (Germany); Appendix A
Netherlands	2004, 2012, 2013 (Germany), 2014, 2016–2018 (Netherlands); Appendix A	fish (Italy), fruits, vegetables (Germany); Appendix A	composition (Germany), pathogenic micro-organisms (Sweden, Netherlands); Appendix A	alert (Germany, Netherlands); Appendix A
Spain	2010, 2012–2018 (Italy); Appendix A	fish (Italy); Appendix A	metals (Italy); Appendix A	alert, information (Italy); Appendix A
United Kingdom	2004 (Italy), 2006–2013, 2016–2018 (United Kingdom); Appendix A	broths, cereals, meat, prepared dishes, soups (United Kingdom), fish (Italy); Appendix A	allergens, foreign bodies (United Kingdom); Appendix A	alert, information (United Kingdom); Appendix A

**Table 7 ijerph-18-01623-t007:** Pearson’s correlation coefficient for amount of food exported within intra EU-trade (in billions of kg, according to the SITC) and number of notifications on food (according to the RASFF).

Countries	Correlation Coefficient	Test Statistic	Critical Statistic
All EU countries	0.82	7.172	2.056
Western EU countries	0.81	4.995	2.160
Eastern EU countries	0.63	2.724	2.201

Critical statistic in two-tailed distribution, confidence level 0.05.

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
