# Peer review of "The Intra-European Union Food Trade with the Relation to the Notifications in the Rapid Alert System for Food and Feed"

_ijerph, 2021, doi:10.3390/ijerph18041623_

Round 1
Reviewer 1 Report
I have to admit that I am confused about this paper. The title and abstract attracted me but the content was rather disappointed. I am not quite against to this paper, but, in that form yes. I suggest to improve your survey and paper form to make it more scientific, logic and readable. There are a lot of remarks, but below I am pointing only five major comments, which support my general review and may be helpful for Author.
- The title suggests, the relation between the intra-EU food trade and food safety will be analysed. It is very interesting and challenging as the problem of food safety (and food security in broaden aspect) is generally liked with the developing countries. So there are not many surveys devoted to food safety (and food security) issues in the developed countries and in the EU especially. That is why I suggest to focus on problems of food safety (as a part of food security), links with international trade, exposure to different risks (e.g. economic, social, environmental) and institutional instruments supporting food safety (e.g. RASFF).
- The intra-EU trade (according countries and products) is important but this problem is topic of many scientific articles (considering regional integrations, competitiveness, etc…) and reports. So in my opinion it can’t be a main aim of this article. Of course, the intra-EU trade should be presented and evaluated as it is part of cluster analysis but the main goal should be concentrated on relation between intra-trade and food safety (RASFF). The scientific problem must be also underline and support by theoretical background.
- You should better explain method you applied and give references. You refer to colours (lines 122-125) although you don’t use them in article. It is not readable and it doesn’t matter if reader knows or doesn’t know this method from technical point of view. Why the period 1999-2018 was chosen? Shorten time period would be better because of new member countries from 2004, activity of RASFF and cluster building. Why do you analyse trade by value and quantity? Is it important considering results? What are the relations between trade value/trade quantity and food safety?
- Presentation of your research results is unacceptable. What are the Figures ….., panels (Lines 157–179, Table 3, Lines 253-281, Table 5), colours (line 272)? It is just unreadable. Some results are obvious “Already preliminary results (Figure S33-S37) indicated that the notified products originated mainly from the same countries that are the main food exporters within the EU.” (line 253-254). If you analyse such a long time period, it would be interesting to evaluate changes (e.g. clusters in 2004/2005 and 2018). Analysis based on averages 1999-2018 can be misleading.
- Discussion and conclusion are not connected with research. There are many new issues (trade protection, Brexit, CAP) which can be partly useful to explain some results but their places in relation trade – food safety are not explained. Limitation of research and further research directions can be indicated.
Author Response
The author would like to thank the Reviewer for all valuable comments to the paper and below he would like to present how he responds to them.
- In the submitted paper the author focused on the economic aspect of the intra-EU food trade (on the base of SITC ) and its connection with notifications in the RASFF. It was not the intention of the author to study social and environmental risks. However, the proposed paper is an extension of the author’s research, as in all previous articles he only referred to the notifications in the RASFF.
- The title and goal of the paper were changed (now they are directly related to the RASFF). When preparing the paper, the author tried to select the most up-to-date and appropriate articles from following databases: Science Direct, Scopus, Springer and Wiley. However, now the theoretical background in the section 1.Introduction was extended with definitions (food security, food safety, risk, hazard, common agricultural policy), which are related mainly with the law.
- In the revised paper in section 2. Data and Methods information on data used in the clusters analysis was added.
The period 1999-2018 was the longest period for which data was available in the Eurostat database at the time of the study, and was chosen to follow changes in intra-EU food trade (Figure 1), taking into account the accession of Central and Eastern European countries to the EU in 2004. Indeed, these countries have benefited on the opening of the European market to their food products. In the revised version of the paper, the author referred to this fact in subsection 3.1 (therefore, new tables (S1-S30) were also added in the Supplementary Material). However, they did not overtake Western European countries (Belgium, France, Germany, Italy, the Netherlands, Spain and the United Kingdom) in intra-EU food trade. Moreover, if the research period was 2005-2018, the same countries would dominate in intra-EU food trade if the same selection criteria were followed, i.e. in each case: imports and exports in kg and euro above the mean value. Noticeable change (decrease) in intra-EU trade in food caused while the economic crisis in 2008-2009, as mentioned in the submitted paper.
Trade by quantity and value was analyzed to show the relationship between them (paragraph before Table 1 in subsection 3.1), while the relations between intra-EU food trade and food safety were shown in subsection 3.2.2 (Table 5 and the text below it) and in the first paragraph of subsection 4.2 (Pearson’s correlation coefficient).
- All figures and panels within particular figures referenced in Table 3 and Table 5 were included in Supplementary Material to the submitted paper. The information about it was placed in section 2. Material and Methods. Table 3 and Table 5 present the results of the research obtained by means of the cluster analysis in a very aggregated way.
The part of the text with the sentence “Already preliminary results…” was rewritten in the revised version.
Indeed, using the mean value can lead to some simplifications, but it is the most frequently used statistical measure (it is also used in the analysis of variance within the multivariate method which is cluster analysis). In these studies, the mean was used to identify the countries with the largest intra-EU food trade over as much as 20 years (the longest possible period with available data at the time of the study, as mentioned in point 3). Therefore, the author believes that the conclusions based on these studies can be considered reliable. Issues related to the accession of Central and Eastern European countries to the EU were presented in point 3.
In the submitted paper, the issues related to the trade protection was raised in subsection 4.1.1 and Brexit and the CAP in subsection 4.2.2 (all of them were also signalled in section 1. Introduction).
In the submitted paper, the author tried to divide the Results into many subsections in order to present them in an accessible and legible way. In section 3 presented the Results and in section 4 presents Discussion. The author believes that subsections 3.1 and 4.1 (relating to intra-EU food trade) and subsections 3.2 and 4.2 (relating to the RASFF) are interrelated.
The first paragraph of the Conclusions concerned the intra-EU trade in food, and the second, the RASFF, and they were related to the research carried out. The rest of the Conclusions did not really follow from the text of the paper and was therefore reformulated or changed. Comments on the limitations of the conducted research were also added and further research directions were indicated.
Reviewer 2 Report
- Paper is interesting, first of all in informative and knowledge spheres concerning the agricultural and food trade flows between countries inside the European Union (in quantity and value) and the notification of food hazards related to traded products during the period 1999-2018.
- The article presents an empirical data and mainly oppinions of other authors, while own assesment and evaluations are rather rare. For instance Author accepts, with no remarks in discussion, an oppinion of Jambor on the lack of competitiveness of Eastern European countries export to other EU countries.There is almost an absence of any theoretical bacground of presented problems; the intra European trade, the food hazards and food safety. The title of the article do not adequately express the contents. At least some definitions of terms and concepts, eg. as alert notifications and informations notifications, food security and safety and links of these terms with an international trade are necessary. A notice on inter-industry trade inside intra european food trade would be justified as well.
- Some readers would expect an assessment on the factors influencing the trade flows, and on describing the main centres/regions of specific food production. The intensive trade exchange between Belgium, Germany, France and Netherlands is highly influenced by location and concentration of production, similarly the trade between south and north countries is influenced by production profile depended on different climate conditions. Links of import and export with a foreign direct investment may be also menthioned as a factors. The value and quantity of trade flows between countries are depended on the size of economy of trading countries. The absolute amount of trade of small/big economies may not represent importance of trade in the national economy in the some way as expressed in relative terms. Trade flows between countries was influenced also by animal diseases (pigs, poultry) in some countries.
- The data concerning trade of European Union countries in the period 1999-2004 and 2004-2018 are not comparable due to the fact that 10 Eastern and South European countries attained membership of the EU Common Market only in 2004. Conclusions based on the analysis of such long period, when Eastern countries was out of the Intra European Trade Borders may be misleading.
Author Response
The author would like to thank the Reviewer for all valuable comments related to the paper and he refers to them below.
- In his articles, the author always tries to clearly separate the results of his own research (section Results) from research or comments of other authors on a given topic (within the section Discussion). The author believes that in the submitted paper the individual subsections of section 3. Results and section 4. Discussion corresponds with each other (3.1 and 4.1 relate to intra-EU food trade, and subsections 3.2 and 4.2 relate to the RASFF), to make the entire paper more accessible and legible to a potential reader. Indeed, the paper presents a critical view of Jambor about the competitiveness of Central and Eastern European within food products, but this opinion is contrasted with the views of several other authors in the next paragraph (subsection 4.1.3). What important, in the revised paper a paragraph on the benefits to the Central and Eastern European countries following their accession to the EU was added (in subsection 3.1).
In the revised paper in section 1. Introduction the terms: intra-EU, food safety, food security, risk, hazard, common agricultural policy were explained. The terms: alert notifications, information notifications were also defined in the submitted paper.
The title and the goal of the paper were changed, i.e. they were directly referred to the notifications in the RASFF.
- In the revised paper in subsection 3.1 signalled the increase of and intra-EU food trade in the total intra-EU trade and also an increase of food production in 1999-2018. Therefore, in Supplementary Material tables (S1-S30) to support these observations were added. Table S1 refers to the intra-EU food trade, Tables S2-S29 relate to intra-EU food trade for particular EU countries and in Table 30 presented food production based on the FAO database as a new source of data in the paper (unfortunately, in the Eurostat database this kind of data was not accessible for the whole examined period).
The author cited research conducted by other authors relating to the specificity of exports of southern European countries resulting from climatic conditions in subsection 4.1.1 in the submitted paper, in turn the hazards related to bacteria and viruses were indicated in subsection 4.2.
- If the analyzed period were to be shortened to 2004-2018 (or 2005-2018), countries with intra-EU food trade above average in all flows considered simultaneously, i.e. imports and exports in kg and euro, would remain the same (and it would be: Belgium, France, Germany, Italy, the Netherlands, Spain and the United Kingdom). However, the adoption of such a long period (1999-2018) made it possible to observe that despite the very significant benefits in the intra-EU food trade that the countries of Central and Eastern Europe gained after their accession to the EU in 2004, they did not manage to achieve an advantage in this trade over Western European countries.It is also worth noting that Romania, Bulgaria and Croatia joined the EU even at a later date.Therefore, the author concluded that the longest possible period (1999-2018) for which data on food trade was available in the Eurostat database (data from the pre-accession period for these countries were also available in this database) was the most suitable for the research.
Reviewer 3 Report
Review repot:
This is a potentially interesting study on food trade within the European Union. Given the current Brexit situation, this study increases the previous knowledge about bilateral trade between the EU and United Kingdom, with important results that aid in understanding potential economic risks. It also provides a picture of trade between EU countries in relation to food safety. This image may help in the design of public policies that improve public health.
The title and abstract are appropriate for the content of the text.
The author has conducted an accurate study.
Author Response
The author would like to thank the Reviewer for the positive review of the paper. In further research, the author will focus on the correlation between food production (according to the Faostat) and food trade within the EU (according to the Eurostat) in the context of both food safety (referring to notification in the RASFF) and food security.
Round 2
Reviewer 1 Report
Thank you for your response and improvement of the paper. I really appreciate it, although I still have some comments and remarks.
- You achieved assumed goal because you presented identified clusters. And the relation between notification RASFF and intra-EU trade is rather treated as the aside effect, what can be understood because it is not the main aim of research. In my opinion that relation would be much more interesting and more scientific goal, especially with identification of determinants. But maybe it is for further research.
- The theoretical background is poor, as I still do not know what economic theory is behind your research. But maybe it is not necessary?
- Methodology is still presented with no references (excluding Statistica handbook, number 34 in Literature). Referring to colours is still understandable in this place (line 129–145). In part 2, you just focus on material. You can assumed that the method you applied is very popular and known widely, so you don’t want to present it (and it is OK), but you should give some references to published literature.
- In line 188–190 or in table 2, I suggest to put the “mean value” that was exceeded.
- Line 195 “it can be assumed that it is directed more to their domestic market than to export”: It can’t be assumed (or you can present statistical data), because those countries can export products to non-EU countries.
- You should place table 3 in part 3.1 or refer to this table in part 3.1.1 and 3.1.2
- Table 4: Change product names (some products, e.g. cereals, eggs start with big letters).
- Presentation of results is still unreadable for me (compare the previous review). Of course I noticed that materials are in Supplementary Material but I did’t get them before. Now having them, it is better but still complicated for readers.
- Lines 474 – 477: “However, there is a high correlation between the amount of food in billions of kg exported from individual countries to other EU countries (according to the SITC) and the number of notifications on food originating from these countries (according to the RASFF, with the exception of those which were from the origin countries).” And lines 536-537: “However, the correlation between the amount of food exported according to the SITC and the number of notifications (excluding those from the origin country) in the RASFF was also high.” Sorry, but I am lost. It is relation between what and what? Try to be more precisely.
- Lines 480-481: “moreover, that products that were most often the subject of intra-EU food trade were usually also most often reported in the RASFF” Such conclusion is obvious. The more you trade, more often you are exposed to higher risk. The question is: why it happens in spite of that the special system (RASFF) is working. What determine the increasing number of notification in intra-EU trade? What are the determinants? Are they the same in old EU countries and for example in the new-EU countries, … etc.
- Lines 525-527: “Three quarters of intra-EU food trade was conducted between Western European countries, i.e. Belgium, France, Germany, Italy, the Netherlands, Spain and the United Kingdom. This indicates a high level of economic integration between these countries.” Are you sure that this mentioned 75% of intra-EU trade means exactly trade among these seven countries? The share of those seven countries in intra-EU trade doesn’t mean that this is the trade among this seven countries.
- Of course, I agree that mean value is popular, but it doesn’t mean it can be used with no reflection. You have such a long research period and the role of particular countries, especially of some new EU countries changed. That could affect clusters. And for example in two shorter time period the clusters could be quite different. But this is only suggestion for future.
Author Response
The author would like to thank the Reviewer for all valuable remarks and comments. Below they are numbered and then quoted using italics while the author's responses are written in regular font. Changes in the current version of the paper are additionally highlighted in yellow.
- Thank you for your response and improvement of the paper. I really appreciate it, although I still have some comments and remarks.
You achieved assumed goal because you presented identified clusters. And the relation between notification RASFF and intra-EU trade is rather treated as the aside effect, what can be understood because it is not the main aim of research. In my opinion that relation would be much more interesting and more scientific goal, especially with identification of determinants. But maybe it is for further research.
The author thanks the Reviewer for this observation. Indeed, the first part of the manuscript title and the first part of its goal are focused on the intra-EU food trade. Most of the research is devoted to this issue. In addition, however, the author tried to link the intra-EU trade in food with notifications in the RASFF. As suggested by the Reviewer in the second review (in comment 10 starting with the phrase "Lines 480-481…"), the author tried to indicate these determinants in this version of the manuscript.
- The theoretical background is poor, as I still do not know what economic theory is behind your research. But maybe it is not necessary?
In order to provide a stronger justification for the premises to take up the topic, the author has included an additional table in section 1 (Table 1). Introduction. This table presents the percentage share of intra-EU and extra-EU food trade. The remaining tables have been renumbered. The author also added a broader definition of "food security" according to the FAO.
- Methodology is still presented with no references (excluding Statistica handbook, number 34 in Literature). Referring to colours is still understandable in this place (line 129–145). In part 2, you just focus on material. You can assumed that the method you applied is very popular and known widely, so you don’t want to present it (and it is OK), but you should give some references to published literature.
Cluster analysis is a fairly frequently used statistical method. However, the author added three new sources in section 2. Materials and Methods to better explain how this method works.
- In line 188–190 or in table 2, I suggest to put the “mean value” that was exceeded.
Following the Reviewer’s suggestion, the author added one row to Table 2 (now Table 3) (at the end of this table) and included the mean value in it.
- Line 195 “it can be assumed that it is directed more to their domestic market than to export”: It can’t be assumed (or you can present statistical data), because those countries can export products to non-EU countries.
The author thanks the Reviewer for this remark. This phrase and the entire sentence was removed. It was actually not a correct assumption. The correctness of this assumption could be assessed only after taking into account the data on the trade of a given EU country with non-EU countries, but such data were not taken from the Eurostat database.
- You should place table 3 in part 3.1 or refer to this table in part 3.1.1 and 3.1.2
The author's intention was that each table was placed on one page (therefore, this table has been moved to subsection 3.1.1). However, as suggested by the Reviewer, in this version of paper version this table (now Table 4) was moved to subsection 3.1.
- Table 4: Change product names (some products, e.g. cereals, eggs start with big letters).
I would like to thank the Reviewer for this remark.The product names in Table 4 (now Table 5) have been corrected to lowercase.The similar changes (for words “export” and “import”) have been made in Table 3 (now Table 4).
- Presentation of results is still unreadable for me (compare the previous review). Of course I noticed that materials are in Supplementary Material but I did’t get them before. Now having them, it is better but still complicated for readers.
Of course, the author understands the Reviewer's comment. The lack of these figures in the paper makes it difficult for the potential reader to read the research results. However, there were so many figures presenting the results of the cluster analysis (several dozen figures, additionally divided into panels) that the author decided to include them in the Supplementary Materials. The author cited each of these figures, preceding their number with the letter "S", pointing to the most significant observations from a given figure, and collecting all this information mainly in Table 3 (now Table 4) and Table 5 (now Table 6). This approach (using Supplementary Materials) has been used by the author in all his previous papers related to the RASFF.
- Lines 474 – 477: “However, there is a high correlation between the amount of food in billions of kg exported from individual countries to other EU countries (according to the SITC) and the number of notifications on food originating from these countries (according to the RASFF, with the exception of those which were from the origin countries).” And lines 536-537: “However, the correlation between the amount of food exported according to the SITC and the number of notifications (excluding those from the origin country) in the RASFF was also high.” Sorry, but I am lost. It is relation between what and what? Try to be more precisely.
Indeed, the sentence in the Conclusions was too simplistic and did not reflect the information from the text of the paper. So it has been changed to a sentence: “However, it was also a high correlation between the amount of food in billions of kg exported from individual countries to other EU countries and the number of notifications on food from these countries reported by other EU countries.”
- Lines 480-481: “moreover, that products that were most often the subject of intra-EU food trade were usually also most often reported in the RASFF” Such conclusion is obvious. The more you trade, more often you are exposed to higher risk. The question is: why it happens in spite of that the special system (RASFF) is working. What determine the increasing number of notification in intra-EU trade? What are the determinants? Are they the same in old EU countries and for example in the new-EU countries, … etc.
The author reformulated the entire first part of subsection 4.2. Table 7 has been added, in which the values of the Pearson’s correlation coefficient for the amount of food exported and the number of notifications in the RASFF (for all EU countries, as well as for Western and Eastern EU countries) are given. Potential factors that can affect food trade and the number of notifications have also been added.
- Lines 525-527: “Three quarters of intra-EU food trade was conducted between Western European countries, i.e. Belgium, France, Germany, Italy, the Netherlands, Spain and the United Kingdom. This indicates a high level of economic integration between these countries.” Are you sure that this mentioned 75% of intra-EU trade means exactly trade among these seven countries? The share of those seven countries in intra-EU trade doesn’t mean that this is the trade among this seven countries.
Indeed, this conclusion was not sufficiently well-formed. As mentioned in the subsection 3.1, the share of intra-EU food trade in total intra-EU trade increased over the period considered (1999-2018), but only to 18% (in volume) and to 11% (in euros). Therefore, the conclusion was refined, with an indication that it relates only to food: “It indicates a high level of economic integration between aforementioned countries in this area.”.
- Of course, I agree that mean value is popular, but it doesn’t mean it can be used with no reflection. You have such a long research period and the role of particular countries, especially of some new EU countries changed. That could affect clusters. And for example in two shorter time period the clusters could be quite different. But this is only suggestion for future.
As the author mentioned in the first responses to the Reviewer's comments, shortening the studied period to 2005-2018 (i.e. after 2004, in which new countries joined the EU) would not change the assumptions regarding the average value. However, it should indeed be noted that the role of Poland in the intra-EU trade in food is increasing. Denmark and Austria also have a noticeable share in this trade. The withdrawal of the United Kingdom from the EU will certainly change a lot in the intra-EU food trade. On the other hand, it will always be an important economic partner for the EU, especially if that country remains in the European Economic Area.